# Position: Causal AI for Personalised Treatments Needs Holistic Approach

## Abstract

Causal AI has strong potential to support person­alised treatments through individualised treatment effect (ITE) estimation. In practice, deployment-time personalised decisions often depend on the *composite* effect of multiple simultaneous inter­ventions, yet many widely used ITE benchmarks, software pipelines, applications, and methodolog­ical work focus on a single intervention or a re­stricted subset. **This position paper argues that** ITE estimation under a restricted intervention set is fundamentally misaligned with personalised treatment decision support, because omitting ac­tionable co-interventions can induce a value gap with respect to the deployment-time objective. We formalise this issue as *omitted intervention bias* and provide a constructive proof that ITE-based decisions under a restricted intervention set can yield *strictly worse achievable outcomes* and *mis­aligned decisions* even when causal effects for the included interventions are correctly identified. We conclude with recommendations on problem formulation, reporting, evaluation, benchmarking, and future research directions to align causal AI with deployment-time personalised treatments.

## 1. Introduction

Causal AI is an umbrella term for an emerging field at the intersection of causality and artificial intelligence (AI), com­bining causal reasoning about interventions with AI's capa­bility to handle complex interactions, high-dimensional vari­ables, and unstructured data, to support principled decision-making beyond purely predictive associations (Kaddour et al., 2022; Chauhan et al., 2025b). At its core is causal inference, which provides a formal framework for reasoning about the effects of interventions, that is, how outcomes would change under different actions (Pearl, 2009; Yao

[1]Anonymous Institution, Anonymous City, Anonymous Region, Anonymous Country. Correspondence to: Anonymous Author <anon.email@domain.com>.

Preliminary work. Under review by the International Conference on Machine Learning (ICML). Do not distribute.

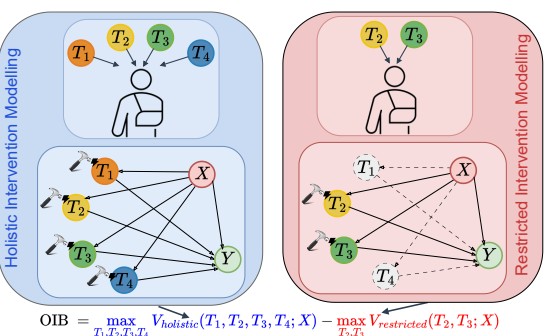

$$\text{OIB} = \max_{T_1,T_2,T_3,T_4} V_{holistic}(T_1,T_2,T_3,T_4; X) - \max_{T_2,T_3} V_{restricted}(T_2,T_3; X)$$

*Potentially suboptimal personalised decisions and worse outcomes from ignoring actionable co-interventions (shown in dotted grey, OIB: omitted intervention bias).*

*Figure 1.* Holistic versus restricted intervention modelling. Here $T_i$ denote interventions, $X$ covariates, $Y$ the outcome, and $V(\cdot; \mathbf{X}) = \mathbb{E}[Y \mid \mathrm{do}(\cdot), \mathbf{X}]$ the value function learned through ITE model.

et al., 2021). In personalised decision-making, this reason­ing is individual-specific and is commonly operationalised via individualised treatment effects (ITEs), which compare expected outcomes for an individual under different inter­ventions (Feuerriegel et al., 2024). Accordingly, ITE es­timation has recently become a central tool in causal AI, often with the implicit assumption that accurate ITE esti­mates, even with a single intervention (or any strict subset), directly support personalised decisions (Curth et al., 2024). Throughout this paper, we use *personalised treatments* to mean prospective *deployment-time* decision support where multiple simultaneous interventions are actionable and the objective is holistic optimisation over those interventions.

---

**Why holistic modelling matters**

**Multi-intervention decisions are routine.** A recent systematic review estimated polypharmacy (typically $\geq 5$ concurrent medicines) at about 37% overall, with rates around 26-40% in older populations (Delara et al., 2022; Bennie et al., 2024).

**Surveys indicate a gap.** Yao et al. (2021) and Cheng et al. (2022) report no causal inference benchmark and no software package for jointly actionable multi-interventions, respectively.

**Risk of decision mismatch and adverse outcomes.** In EGFR-mutated advanced non-small-cell lung cancer, clinicians choose among simultaneous interventions (osimertinib, gefitinib, platinum chemotherapy, pemetrexed, amivantamab, laz­ertinib), yet trials test only selected combinations (Soria et al., 2018; Planchard et al., 2023; Yang et al., 2025); this can misalign deployment-time personalised decisions with achievable outcomes, motivating holistic intervention modelling.

---

**In this paper, we take a position that** the prevailing formu­lation of ITE estimation under a restricted set of interven­tions is fundamentally misaligned with personalised treat­ment decision-making when multiple interventions are ac-

tionable at deployment. In such settings, optimising over a single intervention (or any strict subset) constitutes an *intervention-set misspecification*: the estimand may be well defined and even correctly identified for the included interventions, but it targets a different decision problem than the one faced at deployment because it optimises over fewer decision levers than are available in practice. We formalise the resulting value gap between holistic and restricted objectives as **omitted intervention bias** (refer to Figure 1 for an illustration), and show by construction that this gap can arise. Its consequence is not only a change in effect interpretation, but potentially *misaligned personalised recommendations* and *degraded achieved outcomes* (policy value). We then outline implications for problem formulation, reporting, evaluation, benchmark design, and future research directions. Importantly, we do not argue against single-intervention ITE estimation; our critique targets overclaiming deployment-time personalised decision support when the intervention set used for optimisation is a strict subset of what is actionable in practice.

## 2. Background and Motivation

**Prevalence of restricted intervention modelling.** Recent advances in causal AI, motivated by personalised decision-making, have increasingly focused on ITE estimation methodologies. However, many widely used ITE benchmarks, software pipelines, and methodological work still adopt a single-intervention formulation, typically involving a binary, continuous, or multivalued treatment (Shalit et al., 2017; Dorie et al., 2019; Oprescu et al., 2019; Sharma & Kiciman, 2020; Schröder et al., 2025), even though real-world applications, particularly in healthcare, routinely involve multiple simultaneous interventions, often referred to as *composite interventions* or polypharmacy (Wastesson et al., 2018). A small number of early attempts consider ITEs under multiple simultaneous interventions, but they remain limited in ways that impede realistic deployment, for example, focusing on binary interventions only or requiring separate models per intervention (Qian et al., 2021), continuous interventions only (Schweisthal et al., 2023), or relying on complex hypernetwork-style meta-learning architectures (Chauhan et al., 2025a). In such settings, clinically meaningful interactions are common, including bundle-style care protocols where outcome improvements depend on coordinated combinations rather than isolated actions (Venkatesh et al., 2022), for example the case of lung cancer discussed above (Yang et al., 2025). This makes action-space restriction a practical safety concern, not only a conceptual one, because recommendations can be driven by extrapolation in poorly supported regions of the joint intervention space, which motivates holistic intervention modelling. Table 1 suggests that support for deployment-time personalised treatment under multiple jointly actionable interventions is

still limited in the current literature.

*Table 1.* Alignment of existing work with deployment-time personalised treatment under multiple actionable interventions.

| Reference | Type | Multi | Action | Omit | Inter. | Eval. | Focus | Act.? | Over.? |
|---|---|---|---|---|---|---|---|---|---|
| Shalit et al. (2017) | M | None | Single | N.A. | Add. | TE | E | ✗ | ✗ |
| Dorie et al. (2019) | M | None | Single | N.A. | N/S | TE | E | ✗ | ✗ |
| Oprescu et al. (2019) | M | None | Single | N.A. | N/S | TE* | E | ✗ | ✗ |
| Qian et al. (2021) | M | Bund. | Restr. | B1/2 | Mod. | TE | E | ◖ | ◖ |
| Schweisthal et al. (2023) | M | Bund. | Restr. | B1/2 | Mod. | Value | E | ◖ | ✓ |
| Chauhan et al. (2025a) | M | Bund. | Hol. | N.A. | Mod. | TE | E | ◖ | ✗ |
| Ngufor et al. (2023) | A | Bund.† | Restr. | B2 | Add. | O/U | E | ✗ | ◖ |
| Xu et al. (2023) | A | Bund.† | Restr. | B2 | Add. | O/U | E | ✗ | ◖ |
| Kennedy (2019) | M | None | Single | N.A. | N.A. | Value | D | ✓ | ✓ |
| Athey & Wager (2021) | M | None | Restr. | Uncl. | N/S | Value | D | ✓ | ◖ |
| Murphy (2003) | M | Seq. | Restr. | Uncl. | Mod. | Value | D | ✓ | ◖ |

*Legend:* Type: M=methodology, A=applied. Multi: None=single intervention; Bund.=simultaneous bundle; Seq.=sequential. Action: Single=assumes one actionable; Restr.=restricted; Hol.=toward holistic. Omit: B1/2=B1 or B2; B2=conditioned-as-fixed; N.A.=not applicable; Uncl.=unclear. Inter.: Mod.=modelled; Add.=additive/implicit; N/S=not stated. Eval.: TE=treatment effect metrics; Value=policy value/regret; O/U=other/unclear; TE*: varies by use. Focus: D=decision support, E=effect estimation. Act.? actionability declaration. Over.? overlap/positivity addressed. †Bundle present in data but not optimised over.

Lack of holistic intervention modelling is evident in many applied ITE studies, which operationalise personalisation by focusing on one intervention while ignoring or treating others as covariates. In medicine, examples include heterogeneous effect estimation for a focal drug choice, for instance in anticoagulation decisions for atrial fibrillation (Xu et al., 2023), or in counterfactual prediction under dynamic treatment strategies evaluated on ICU datasets (Xiong et al., 2024). In economics and public policy, prominent work estimates heterogeneous impacts of a single programme (for example, youth employment interventions) while leaving other concurrent policy actions outside the intervention set (Davis & Heller, 2017). In marketing, uplift modelling commonly estimates the incremental impact of a single campaign at the individual level, despite multiple actionable interventions (Radcliffe & Surry, 2011; Goldenberg et al., 2025). These formulations can be appropriate under alternative objectives (see Section 5), but they also illustrate a recurring pattern: estimands and evaluation protocols privilege single-intervention effects even when deployment-time decisions can involve jointly selecting multiple actionable, interacting interventions.

**Why the field defaults to restricted intervention modelling.** The prevalence of single-intervention ITE formulations is driven by a combination of structural, methodological, application, and benchmarking constraints: *(i)* Data scarcity and practical violations of positivity become severe when moving beyond single interventions, as many clinically or operationally plausible treatment combinations are rare or unobserved, undermining identifiability and yielding unstable estimation (Petersen et al., 2012; D'Amour et al., 2021). *(ii)* The joint intervention space grows combinatorially with the number of interventions, dosages, and timings, quickly overwhelming standard modelling and evaluation pipelines; this challenge has been recognised in

multi-treatment uplift settings (Zhao et al., 2017; Wei et al., 2024). *(iii)* Methodological support for multiple simultaneous interventions remains limited, particularly beyond binary combinations (Qian et al., 2021). *(iv)* The applications are mostly aimed at retrospective effect estimation rather than deployment-time decision support (Xu et al., 2023; Ngufor et al., 2023). *(v)* Finally, existing libraries and benchmarks are limited to single interventions (Dorie et al., 2019; Sharma & Kiciman, 2020; Machlanski et al., 2025).

## 3. Preliminaries

**Notation.** Random variables use uppercase (e.g. $T_1$) and realisations lowercase (e.g. $T_1 = t_1$); vectors are bold (e.g. $\mathbf{X}$, $\mathbf{X} = \mathbf{x}$).

Let $\mathbf{X} \in \mathcal{X}$ denote observed pre-intervention covariates for an individual, $Y$ an outcome of interest, and $T_1, \ldots, T_K$ a set of actionable[1] interventions, where each $T_k \in \mathcal{T}_k$ may be binary, categorical, or continuous. For each individual and each joint intervention $(t_1, \ldots, t_K)$, let $Y(t_1, \ldots, t_K)$ denote the potential outcome that would be observed if the corresponding interventions were assigned. We adopt the following standard assumptions to support identification from observational data (Imbens & Rubin, 2015; Hernan & Robins, 2025).

**Assumption 3.1** (Consistency). *If an individual receives interventions $(T_1, \ldots, T_K) = (t_1, \ldots, t_K)$, then the observed outcome satisfies $Y = Y(t_1, \ldots, t_K)$.*

**Assumption 3.2** (Unconfoundedness). *Conditional on observed covariates $\mathbf{X}$, intervention assignment is independent of potential outcomes, that is, $Y(t_1, \ldots, t_K) \perp (T_1, \ldots, T_K) \mid \mathbf{X}$ for all $(t_1, \ldots, t_K)$.*

**Assumption 3.3** (Positivity). *For all $(t_1, \ldots, t_K)$ and all $\mathbf{x} \in \mathcal{X}, 0 < \mathbb{P}(T_1 = t_1, \ldots, T_K = t_K \mid \mathbf{X} = \mathbf{x}) < 1$.*

**Definition 3.4** (Individualised Treatment Effect (ITE)). *The individualised treatment effect (ITE), also referred to as the conditional average treatment effect (CATE) or heterogeneous treatment effect (HTE) in machine learning literature, is defined as the contrast between potential outcomes under two intervention assignments for an individual with covariates $\mathbf{X} = \mathbf{x}$ (Curth & van der Schaar, 2021a; Melnychuk et al., 2025a). For example, for intervention $T_1$,*

$$\text{ITE}_{T_1}(\mathbf{x}; t_1, t_1') := \mathbb{E}[Y(t_1) - Y(t_1') \mid \mathbf{X} = \mathbf{x}],$$

*where $Y(t_1)$ denotes the potential outcome under the intervention $\text{do}(T_1 = t_1)$ with all other interventions left unspecified. Importantly, ITEs are defined with respect to a*

---

[1]We say an intervention is *actionable* if it can be chosen or influenced by the decision-maker at the time the decision is made (deployment), rather than being fixed by protocol or determined by an external process.

specified set of intervention variables (e.g., here only $T_1$), and their interpretation depends fundamentally on which interventions are treated as actionable in the estimand.

## 4. Why Holistic Modelling Is Necessary

### 4.1. Decision target and definitions

**Motivation.** When multiple interventions are jointly actionable at deployment, personalised treatment is inherently a *joint* optimisation problem over the actionable intervention set. We call an approach *holistic* if the ITE model (and the induced decision rule) is defined over this full set; many existing formulations instead optimise over a restricted subset, inducing intervention-set misspecification and potentially degraded achieved outcomes even when effects for the included interventions are correctly identified.

To formalise the personalised treatment decision target, we define an individual-level *value function* mapping covariates and joint intervention assignments to the expected outcome:

$$V(t_1, \ldots, t_K; \mathbf{x}) := \mathbb{E}[Y \mid \text{do}(t_1, \ldots, t_K), \mathbf{X} = \mathbf{x}].$$

In practice, $V$ is unknown and is approximated by an ITE model that estimates conditional potential outcomes.

---

**Definition: Personalised treatment**

Assume $K$ interventions are jointly actionable at deployment. A personalised treatment decision selects a joint assignment $(t_1, \ldots, t_K)$ for an individual $\mathbf{X} = \mathbf{x}$ to optimise the expected outcome over the *full actionable intervention set*:

$$(t_1^\star, \ldots, t_K^\star) \in \arg \max_{t_1, \ldots, t_K} V(t_1, \ldots, t_K; \mathbf{x}).$$

---

In practice, many analyses restrict attention to only a subset of the available interventions. For example, an ITE method may model only $T_1$ while (i) ignoring or (ii) treating other interventions as covariates, leading to a restricted objective of the form

$$\max_{t_1} V(t_1; \mathbf{x}), \quad \text{and} \quad \max_{t_1} V(t_1; t_2, \ldots, t_K, \mathbf{x}),$$

respectively, where omitted interventions do not appear as optimisation variables.

---

**Definition: Omitted intervention bias (OIB)**

OIB arises when multiple interventions are actionable at deployment, but the decision rule optimises over only a strict subset, treating the remaining actionable interventions as part of the environment (for example, ignored or treated as covariates). Formally, let $T_1, \ldots, T_K$ be actionable, but optimise only over $T_1, \ldots, T_m$ with $m < K$. For an individual with covariates $\mathbf{x}$, a value-based characterisation (when omitted interventions are ignored) is

$$\max_{t_1, \ldots, t_K} V(t_1, \ldots, t_K; \mathbf{x}) - \max_{t_1, \ldots, t_m} V(t_1, \ldots, t_m; \mathbf{x}).$$

---

This bias is not an estimation artefact due to unobserved confounding or outcome-model misspecification. Instead, it reflects a mismatch between the interventions included in the estimand and those required by the personalised decision problem, even when causal effects are correctly identified for the included interventions. Although we introduce

OIB in the context of ITE-based personalised decision support, related concerns exist, for example in econometrics (Mueller-Smith, 2015); contrasts with adjacent literatures are discussed in Appendix D.5.

## 4.2. Main theorem and proof

**Objective.** The goal of this subsection is to make the mismatch between holistic and restricted intervention modelling explicit. Using a causal model consistent with the directed acyclic graph in Figure 2, we derive the value targets induced by (i) holistic joint optimisation and (ii) restricted optimisation. We show that these targets generally differ, which can yield different recommendations and a value gap, thereby motivating holistic intervention modelling.

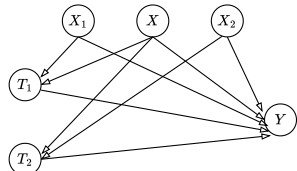

*Figure 2.* Directed acyclic graph for the causal model in Eq. (1).

**Theorem 4.1** (Omitted intervention bias). *For a personalised treatment problem with multiple actionable (composite) interventions, if decisions are supported by an ITE estimand that models a strict subset of interventions, then omitted intervention bias can arise even when causal effects for the included interventions are correctly identified.*

**Corollary 4.2** (Decision mismatch is possible). *Under the conditions of Theorem 4.1, there exist settings and individuals for which the interventions selected under the restricted optimisation differ from a jointly optimal intervention under the holistic optimisation.*

**Proof.** We prove the claim by construction using a structural causal model (SCM) consistent with the directed acyclic graph in Figure 2 (Pearl, 2009). Consider two interventions $T_1, T_2 \in \{0, 1\}$ (that is, $K = 2$) that are jointly actionable at deployment, an outcome $Y$, a common confounder $\mathbf{X}$, and intervention-specific confounders $\mathbf{X}_1$ and $\mathbf{X}_2$ as depicted in Figure 2. We use the following structural equation:

$$Y := \alpha_1 T_1 + \alpha_2 T_2 + \beta_0 g(\mathbf{X}) T_1 T_2 + \beta_1 g_1(\mathbf{X}_1) \\ + \beta_2 g_2(\mathbf{X}_2) + \varepsilon, \quad (1)$$

where $g, g_1, g_2$ are functions, mapping vectors to scalars (e.g., a clinical score) and $\mathbb{E}[\varepsilon \mid \mathbf{X}, \mathbf{X}_1, \mathbf{X}_2, T_1, T_2] = 0$. We distinguish the following scenarios:

● **Scenario A**. *Holistic intervention modelling.* Under the joint intervention $\mathrm{do}(T_1 = t_1, T_2 = t_2)$, SCM semantics replace the structural assignments for $T_1, T_2$ by constants,

yielding

$$Y = \alpha_1 t_1 + \alpha_2 t_2 + \beta_0 g(\mathbf{X}) t_1 t_2 + \beta_1 g_1(\mathbf{X}_1) \\ + \beta_2 g_2(\mathbf{X}_2) + \varepsilon.$$

Therefore, for an individual with $(\mathbf{X}, \mathbf{X}_1, \mathbf{X}_2) = (\mathbf{x}, \mathbf{x}_1, \mathbf{x}_2)$, the expected outcome is

$$V(t_1, t_2; \mathbf{x}, \mathbf{x}_1, \mathbf{x}_2) \\ := \mathbb{E}[Y \mid \mathrm{do}(t_1, t_2), \mathbf{X} = \mathbf{x}, \mathbf{X}_1 = \mathbf{x}_1, \mathbf{X}_2 = \mathbf{x}_2] \\ = \alpha_1 t_1 + \alpha_2 t_2 + \beta_0 g(\mathbf{x}) t_1 t_2 + \beta_1 g_1(\mathbf{x}_1) + \beta_2 g_2(\mathbf{x}_2). \quad (2)$$

The holistic personalised decision and outcome is

$$(t_1^\star, t_2^\star) \quad \in \quad \arg\max_{t_1, t_2} V(t_1, t_2; \mathbf{x}, \mathbf{x}_1, \mathbf{x}_2), \quad (3)$$

$$V(t_1^\star, t_2^\star; \mathbf{x}, \mathbf{x}_1, \mathbf{x}_2) = \alpha_1 t_1^\star + \alpha_2 t_2^\star + \beta_0 g(\mathbf{x}) t_1^\star t_2^\star \\ + \beta_1 g_1(\mathbf{x}_1) + \beta_2 g_2(\mathbf{x}_2). \quad (4)$$

Next, we derive the corresponding objectives under restricted intervention modelling, considering two common practices: **(B1)** the omitted intervention is ignored as background and not optimised over, and **(B2)** the omitted intervention is treated as a covariate and then treated as fixed when optimising.

● **Scenario B1**. *Restricted intervention modelling when the omitted intervention is ignored as background.* A common subset-based formulation treats $T_1$ as the only intervention being optimised, so Eq. (1) becomes

$$Y = \alpha_1 t_1 + \alpha_2 T_2 + \beta_0 g(\mathbf{X}) t_1 T_2 + \beta_1 g_1(\mathbf{X}_1) + \beta_2 g_2(\mathbf{X}_2) + \varepsilon.$$

For an individual with $(\mathbf{X}, \mathbf{X}_1, \mathbf{X}_2) = (\mathbf{x}, \mathbf{x}_1, \mathbf{x}_2)$,

$$V(t_1; \mathbf{x}, \mathbf{x}_1, \mathbf{x}_2) := \mathbb{E}[Y \mid \mathrm{do}(t_1), \mathbf{X} = \mathbf{x}, \mathbf{X}_1 = \mathbf{x}_1, \mathbf{X}_2 = \mathbf{x}_2], \\ = \alpha_1 t_1 + \beta_1 g_1(\mathbf{x}_1) + \beta_2 g_2(\mathbf{x}_2) \\ + \alpha_2 \mathbb{E}[T_2 \mid \mathrm{do}(t_1), \mathbf{X} = \mathbf{x}, \mathbf{X}_1 = \mathbf{x}_1, \mathbf{X}_2 = \mathbf{x}_2] \\ + \beta_0 g(\mathbf{x}) t_1 \mathbb{E}[T_2 \mid \mathrm{do}(t_1), \mathbf{X} = \mathbf{x}, \mathbf{X}_1 = \mathbf{x}_1, \mathbf{X}_2 = \mathbf{x}_2].$$

Let $m_{t_1}(\mathbf{x}, \mathbf{x}_1, \mathbf{x}_2) \\ := \mathbb{E}[T_2 \mid \mathrm{do}(t_1), \mathbf{X} = \mathbf{x}, \mathbf{X}_1 = \mathbf{x}_1, \mathbf{X}_2 = \mathbf{x}_2].$

Then $V(t_1; \mathbf{x}, \mathbf{x}_1, \mathbf{x}_2)$
$$= \alpha_1 t_1 + \beta_1 g_1(\mathbf{x}_1) + \beta_2 g_2(\mathbf{x}_2) + \alpha_2 m_{t_1}(\mathbf{x}, \mathbf{x}_1, \mathbf{x}_2) \\ + \beta_0 g(\mathbf{x}) t_1 m_{t_1}(\mathbf{x}, \mathbf{x}_1, \mathbf{x}_2). \quad (5)$$

The subset-based personalised decision and outcome is

$$t_1^\dagger \in \arg\max_{t_1} V(t_1; \mathbf{x}, \mathbf{x}_1, \mathbf{x}_2), \quad (6)$$

$$V(t_1^\dagger; \mathbf{x}, \mathbf{x}_1, \mathbf{x}_2) = \alpha_1 t_1^\dagger + \beta_1 g_1(\mathbf{x}_1) + \beta_2 g_2(\mathbf{x}_2)$$
$$+ \alpha_2 m_{t_1}(\mathbf{x}, \mathbf{x}_1, \mathbf{x}_2) + \beta_0 g(\mathbf{x}) t_1^\dagger m_{t_1}(\mathbf{x}, \mathbf{x}_1, \mathbf{x}_2). \quad (7)$$

Eqs. (3) and (6) define different decision objectives: the holistic objective optimises the joint-intervention value $V(t_1, t_2; \cdot)$, whereas the restricted objective optimises a background-policy-induced quantity that depends on $m_{t_1}(\mathbf{x}, \mathbf{x}_1, \mathbf{x}_2)$. Because $m_{t_1}(\cdot)$ can vary with $t_1$ and need not coincide with the optimised choice of $t_2$, the induced maximiser $t_1^\dagger$ can differ from the jointly optimal $t_1^\star$, and the achieved value can be strictly smaller than the holistic optimum. A concrete instantiation yielding both decision mismatch and a strict value gap is given in Example 4.3.

● **Scenario B2**. *Restricted intervention modelling when the omitted intervention is conditioned on (treated as fixed context).* A second common practice conditions on the co-intervention $T_2$ and then optimises only over $T_1$, so Eq. (1) becomes

$$Y = \alpha_1 t_1 + \alpha_2 T_2 + \beta_0 g(\mathbf{X}) t_1 T_2 + \beta_1 g_1(\mathbf{X}_1) + \beta_2 g_2(\mathbf{X}_2) + \varepsilon.$$

For an individual with $(T_2, \mathbf{X}, \mathbf{X}_1, \mathbf{X}_2) = (t_2, \mathbf{x}, \mathbf{x}_1, \mathbf{x}_2)$,

$$V(t_1; t_2, \mathbf{x}, \mathbf{x}_1, \mathbf{x}_2)$$
$$:= \mathbb{E}[Y \mid \mathrm{do}(t_1), T_2 = t_2, \mathbf{X} = \mathbf{x}, \mathbf{X}_1 = \mathbf{x}_1, \mathbf{X}_2 = \mathbf{x}_2],$$
$$= \alpha_1 t_1 + \alpha_2 t_2 + \beta_0 g(\mathbf{x}) t_1 t_2 + \beta_1 g_1(\mathbf{x}_1) + \beta_2 g_2(\mathbf{x}_2). \quad (8)$$

The subset-based personalised decision and outcome is

$$t_1^\dagger \in \arg\max_{t_1} V(t_1; t_2, \mathbf{x}, \mathbf{x}_1, \mathbf{x}_2), \quad (9)$$

$$V(t_1^\dagger; t_2, \mathbf{x}, \mathbf{x}_1, \mathbf{x}_2)$$
$$= \alpha_1 t_1^\dagger + \alpha_2 t_2 + \beta_0 g(\mathbf{x}) t_1^\dagger t_2 + \beta_1 g_1(\mathbf{x}_1) + \beta_2 g_2(\mathbf{x}_2). \quad (10)$$

Crucially, when $T_2$ is actionable, conditioning on $t_2$ does not recover the holistic objective because $t_2$ itself should be optimised jointly rather than treated as fixed. A concrete instantiation yielding both decision mismatch and a strict value gap is given in Example B.1. $\qquad \square$

**An alternative derivation** of the same result using the potential outcomes framework under standard identification assumptions (consistency, unconfoundedness, and positivity) is provided in **Appendix A**.

We also corroborate our results with a **simulation (Appendix C)**, showing that a restricted ITE learner policy can exhibit non-trivial decision mismatch and a positive estimated OIB (value gap) relative to holistic joint optimisation, with the gap increasing as intervention interactions strengthen and overlap deteriorates.

*Example* 4.3 (Scenario B1). Fix $(\mathbf{x}, \mathbf{x}_1, \mathbf{x}_2)$ such that $g(\mathbf{x}) = 1$ and $g_1(\mathbf{x}_1) = g_2(\mathbf{x}_2) = 0$. Choose parameters

$$\alpha_1 = -0.2, \quad \alpha_2 = 0, \quad \beta_0 = 2.5, \quad \beta_1 = \beta_2 = 0.$$

*Scenario A (holistic).* From Eq. (2),

$$V(t_1, t_2; \mathbf{x}, \mathbf{x}_1, \mathbf{x}_2) = -0.2\,t_1 + 2.5\,t_1 t_2.$$

Enumerating $(t_1, t_2) \in \{0, 1\}^2$ gives

$$V(0, 0) = 0, V(0, 1) = 0, V(1, 0) = -0.2, V(1, 1) = 2.3,$$

so $(t_1^\star, t_2^\star) = (1, 1)$ and $\max_{t_1, t_2} V(t_1, t_2; \cdot) = 2.3$.

*Scenario B1 (restricted).* Assume $m_0(\mathbf{x}, \mathbf{x}_1, \mathbf{x}_2) = m_1(\mathbf{x}, \mathbf{x}_1, \mathbf{x}_2) = p$ with $p = 0.05$. With $\alpha_2 = 0$ and $\beta_1 = \beta_2 = 0$, Eq. (5) reduces to

$$V(t_1; \mathbf{x}, \mathbf{x}_1, \mathbf{x}_2) = -0.2\,t_1 + 2.5\,t_1 p.$$

Thus, $V(0) = 0, \; V(1) = -0.2 + 2.5 \times 0.05 = -0.075,$

so $t_1^\dagger = 0$ and $\max_{t_1} V(t_1; \cdot) = 0$, and the value gap is $2.3 - 0 = 2.3$. Thus, restricted modelling leads to a decision mismatch and worse outcomes.

## 5. Alternative Views

### 5.1. Single-intervention ITE is sufficient because only one intervention is actionable

A common perspective is that the decision problem genuinely involves a single actionable lever, for example, choose whether to administer a drug. This framing is common in clinical practice, partly for interpretability and perceived reliability (Ngufor et al., 2023). In such a case, focusing on a binary or low-cardinality treatment and estimating ITE is well aligned with the decision problem, and a large methodological literature supports this workflow. Representative examples include meta-learners (Künzel et al., 2019; Melnychuk et al., 2025a), causal trees and forests (Athey & Imbens, 2016; Wager & Athey, 2018), and representation learning based learners (Melnychuk et al., 2022; Wen et al., 2025; Lacombe & Sebag, 2025). This alternative view is compatible with our argument when the omitted interventions are not actionable at the point of deployment, or when the intended use is descriptive (learning about one intervention) rather than prescriptive (optimising over all actionable interventions). Our critique applies when omitted interventions are actionable and should be jointly selected to optimise the outcome.

### 5.2. The target is an incremental effect not joint optimisation

Many applied studies and deployed systems seek an incremental effect of one focal intervention relative to a baseline

or prevailing practice, rather than the best joint combination of all actions. In marketing, uplift modelling is often framed as estimating the incremental impact of a single campaign or contact policy versus control, with decision rules that rank individuals by estimated lift (Radcliffe & Surry, 2011). Incremental propensity score interventions are another example, motivated in part by practical challenges such as limited overlap (Kennedy, 2019). These perspectives are scientifically coherent, but they correspond to a different decision target than the holistic personalised objective in Section 4. The risk arises when an incremental estimand is implicitly interpreted as supporting holistic personalised treatment decisions in settings where multiple interventions are actionable and interact.

### 5.3. Effect estimation is the primary goal, not outcome maximisation

Many studies are conducted with the primary aim of estimating the effectiveness of a focal intervention, while motivating the work in terms of personalised treatment for that intervention. Under this view, progress is framed as improving ITE estimation accuracy, and decision-making is treated as downstream (Curth & van der Schaar, 2021a; Xiong et al., 2024). Representative applied work includes studies of individualised effects of oxygen in critically ill patients (Buell et al., 2024), and related retrospective analyses of heterogeneous treatment effects in real clinical settings (Ngufor et al., 2023). This alternative view is credible and often valuable, e.g., when the scientific objective is to characterise heterogeneity in causal effects. However, when the goal is deployment-time personalised decision support with multiple actionable interventions, the relevant object is the outcome achieved under the induced decision procedure, and ITE accuracy for a restricted intervention set can be insufficient for guiding optimal decisions.

### 5.4. Controlling for co-interventions as covariates addresses the issue

Another viewpoint is that co-interventions can be incorporated into the covariate set and the analysis can proceed by estimating conditional effects for the focal intervention. This is common in observational studies, where adjustment for observed variables is used to support identification under assumptions such as unconfoundedness, positivity, and consistency (Hernan & Robins, 2025). In machine learning practice, this often corresponds to estimating ITE for one intervention while treating other actions as features in the regression or representation. For example, Ngufor et al. (2023) treated beta-blocker use as a covariate while focusing on oral anticoagulation. This approach can yield valid causal contrasts for the included intervention under its stated assumptions, but it does not generally solve the *personalised decision* problem when the omitted interventions

are actionable. Section 4 makes this precise via Scenario B2.

### 5.5. Holistic modelling is infeasible due to positivity and data scarcity

A practical counterargument is that holistic optimisation over many interventions is often infeasible with finite observational data, because the joint action space expands with the number of interventions, dosages, and timings, making many combinations rare and creating severe positivity and overlap challenges that are further exacerbated in high-dimensional covariate settings (Kennedy, 2019; D'Amour et al., 2021; Petersen et al., 2012). From this viewpoint, restricting the intervention set is a pragmatic path, which can be useful for *hypothesis generation*, with validation deferred to confirmatory studies (Feuerriegel et al., 2024). Relatedly, overlap-focused techniques (for example, overlap weights or overlap regularisation) (Hess et al., 2025; Melnychuk et al., 2025b) can improve estimation stability within regions of adequate support, but they do not resolve the core issue here: restricting the actionable intervention set changes the decision objective. We agree that feasibility constraints are real and often binding. However, they do not eliminate the conceptual mismatch highlighted in Section 4.

### 5.6. Interactions are negligible, so separate optimisation is sufficient

Another credible view is that intervention interactions are weak or absent, so the outcome response surface is approximately additive across interventions. Under strong additivity conditions, selecting each intervention based on its marginal effect can coincide with the joint optimum. For example, additivity is sometimes assumed implicitly by fitting models without interaction terms when analysing concurrent chemoradiotherapy in curative cancer treatment (Baumann et al., 2020) or computer assisted learning interventions (Eames et al., 2026). While additivity is a sufficient condition for avoiding OIB, it is a substantive assumption about the deployed setting, and it is likely to be violated in domains where multiple concurrent actions jointly determine outcomes (for example polypharmacy). When interactions are present, Section 4 shows that optimising a subset-based objective can yield misaligned decisions and worse achievable outcomes.

### 5.7. Policy learning and dynamic treatment regimes address complex decision optimisation

Policy learning and dynamic treatment regimes (DTRs) are designed to optimise outcomes under decision rules. DTRs formalise sequential decision-making and aim to learn regimes that maximise expected outcomes (Murphy, 2003; Zhao et al., 2015). Related work in econometrics formulates policy learning as welfare maximisation from observational

data (Athey & Wager, 2021). These frameworks are highly relevant to our emphasis on achieved outcomes, but they do not remove the concern raised here. If the learned regime or policy excludes interventions that are actionable and relevant at deployment, then it still solves a restricted optimisation problem. In this sense, OIB is an action-space misspecification issue that can arise in ITE-based decision support as well as in DTR and policy learning formulations.

---

**Takeaway from alternative views**

The alternative views above clarify circumstances where restricted-intervention ITE estimation can be appropriate. **Our position concerns the common but often implicit shift from these settings to claims about holistic personalised treatment decision-making in environments with multiple actionable, interacting interventions. This shift can encourage practitioners to deploy off-the-shelf single-intervention ITE models for multi-intervention decisions, risking harmful recommendations and adverse outcomes when actionability and interactions are ignored.** Accordingly, responsible practice requires making actionability assumptions explicit and evaluating models against the decision objective they are intended to support, which we turn to next.

---

# 6. Call for Action

## 6.1. Clarify the study objective: retrospective analysis versus deployment-time decision support

A recurring source of confusion in applied work is slippage between two distinct objectives. A *retrospective (backward-looking)* objective aims to explain historical data by estimating the effect of a focal intervention (possibly with heterogeneity), often motivated by personalised treatments, and this is where much of the applied literature lies (Ngufor et al., 2023; Buell et al., 2024). A *deployment-time (forward-looking)* objective aims to support prospective decisions for new individuals by selecting interventions to optimise outcomes, which aligns more naturally with policy learning and welfare-based evaluation perspectives (Athey & Wager, 2021; Kitagawa & Tetenov, 2018). Therefore, authors should make explicit whether their primary intent is retrospective effect characterisation (even if motivated by personalisation) or deployment-time decision support, and align estimands, claims, and evaluation accordingly.

## 6.2. Specify the actionable intervention set and scope

The central practical implication is that authors should explicitly document the decision context the estimand is intended to support. **(i) Declare the intervention set.** State which interventions are assumed actionable at decision time, and which relevant variables are treated as non-actionable (fixed by protocol, determined externally, or outside scope). When interventions admit multiple versions or bundled implementations, articulate what each intervention means (and what is held fixed) to support causal interpretation and transportability (Hernán & VanderWeele, 2011). This clarifies whether the intended objective is single-action decision support, incremental effects under usual care, or holistic joint

optimisation. **(ii) State validity conditions.** When modelling only a subset of interventions, state the conditions under which the resulting decision procedure is intended to be used, and when it should *not* be used. This parallels standard practice in model documentation and disclosure, but with the actionable intervention set as the key validity condition (Mitchell et al., 2019). **(iii) Specify the objective.** Specify the outcome being optimised (single outcome, composite outcome, or constrained objective) and whether optimisation is over a joint intervention choice. Without this, 'personalisation' claims are hard to interpret.

## 6.3. Evaluate decision quality, not only ITE accuracy

ITEs are intervention-centred objects. Personalised treatment decision support, by contrast, is outcome-centred. When the intended use is deployment-time decision support (rather than retrospective effect characterisation), evaluation should therefore prioritise *decision quality*. Concretely, the most relevant target is the expected outcome achieved under the induced decision procedure (policy value) and its gap to an appropriate reference (value loss or regret), similar to policy learning literature (Kitagawa & Tetenov, 2018; Athey & Wager, 2021). We therefore recommend that papers *claiming or motivating deployment-time personalised decision support* report *(i)* a decision-focused evaluation when feasible, and *(ii)* explicit alignment between the evaluated estimand and the deployment intervention set, and, *(iii)* when restriction is unavoidable, authors should include sensitivity checks that probe robustness to OIB.

## 6.4. Build benchmarks and software pipelines that reflect joint actionability

Current benchmarks and software pipelines structurally encourage the single-intervention default. This can incentivise broad claims about personalisation that exceed what the benchmark represents. For example, widely used semi-synthetic ITE benchmarks (IHDP, Jobs, Twins, ACIC, News) and widely used software pipelines (DoWhy, EconML, CausalML, causalToolbox) focus on single-interventions (Yao et al., 2021; Cheng et al., 2022). Recent critiques argue that such benchmark design choices can obscure the assumptions and failure modes that matter for deployment claims (Curth et al., 2021). To better align research with deployment needs: (**(i) Record co-interventions.**) Where feasible, record major co-interventions, and document which interventions are considered actionable in the intended decision context. **(ii) Create multi-action benchmark tasks.** Create tasks that require joint optimisation over at least two interventions, including sparse-support settings where actionability must be stated explicitly (Dorie et al., 2019). **(iii) Extend libraries and pipelines.** Provide reference implementations, APIs, and evaluation utilities for multi-intervention mod-

elling and value-based evaluation, to avoid packaging single-intervention defaults as decision-support tools.

### 6.5. Report overlap and implement safety checks

Limited overlap and practical positivity violations are often the binding constraint for holistic decision support with multiple interventions, because many joint assignments are rare and naïve optimisation can rely on extrapolation and yield unstable recommendations (Petersen et al., 2012; D'Amour et al., 2021). We therefore recommend reporting joint-action support diagnostics (for example, frequencies of joint assignments and overlap/propensity diagnostics), using overlap-focused weighting or regularisation where appropriate (with its implied target shift) (Hess et al., 2025; Melnychuk et al., 2025b), and enforcing explicit safety rules such as abstention, restricted recommendation sets, or deferral to standard of care when support is insufficient.

### 6.6. Monitor deployment drift in the intervention set

In practice, the actionable intervention set is not static: new drugs and protocols are introduced, resource constraints change, and care pathways evolve. This dynamism increases the risk that a model trained or evaluated under one actionable set is deployed under another. Treat the intervention-set declaration as a living artefact: deployment owners should periodically reassess what is actionable, document changes, and re-evaluate the risk of OIB when the decision context shifts. More generally, this reflects standard operational concerns in deployed machine learning systems, including distributional change and ongoing maintenance burdens (Gama et al., 2014; Sculley et al., 2015).

### 6.7. Future research directions

This paper does not propose a complete algorithmic solution for holistic ITE estimation; instead, it motivates the following research directions to align causal AI with deployment-time personalised treatment decisions: **(i) Composite-intervention ITE methodology.** Develop estimands and estimators for ITEs over *joint* intervention choices (including bundles, dosages, and timing), beyond single-intervention and strict-subset formulations. **(ii) Structured action representations.** Represent multi-action choices compositionally (for example, bundled or factorised structures) rather than enumerating combinations, aligning with concerns about compound or multi-version interventions (Hernán & VanderWeele, 2011). **(iii) Meta-learning for data scarcity.** Develop information-sharing and meta-learning strategies that borrow strength across related intervention combinations to mitigate the combinatorial sparsity induced by composite interventions (Chauhan et al., 2025a). **(iv) Positivity and overlap-aware objectives, evaluation, and safety.** Design methods that account explicitly for limited joint-action

support (for example, overlap-aware weighting/regularisation, conservative or support-constrained optimisation, and uncertainty-aware objectives), and couple them with abstention/deferral or restricted recommendation sets to avoid extrapolative decisions (Petersen et al., 2012; D'Amour et al., 2021). **(v) Sensitivity analysis.** Develop sensitivity analyses that quantify how recommendations and policy value change under plausible misspecification of omitted interventions and interactions (Frauen et al., 2024). **(vi) Links to policy learning and DTRs.** Connect holistic intervention modelling to policy learning and dynamic treatment regime frameworks (Athey & Wager, 2021; Murphy, 2003), while keeping the actionable intervention set explicit. **(vii) Benchmark and library infrastructure.** Develop benchmarks and open-source libraries that natively support multi-intervention estimands, overlap diagnostics, and value-based evaluation under joint optimisation (Section 6.4).

We summarise the reporting recommendations below.

| Recommended Reporting Template |
| --- |
| **Study objective.** Deployment-time decision support *or* retrospective effect characterisation (state one). |
| **Actionable interventions at deployment.** $\mathcal{A} = \{T_1, \ldots, T_K\}$. |
| **Modelled interventions.** $\mathcal{M} = \{T_1, \ldots, T_m\} \subset \mathcal{A}$ (if restricted, state $m < K$). |
| **Handling of omitted actionable interventions.** *B1: ignored* (usual care) *or B2: fixed context* (state which and why). |
| **Optimisation objective.** Outcome to optimise (single or composite, constraints) and whether optimisation is joint over $\mathcal{A}$ or restricted to $\mathcal{M}$. |
| **Scope and limitations.** If $\mathcal{M} \subset \mathcal{A}$, state that conclusions apply only to $\mathcal{M}$. |
| **Primary evaluation for decision support.** Policy value (or regret/value loss) relative to a reference policy; report uncertainty when feasible. |
| **Support and safety checks.** Overlap and positivity diagnostics; rule for abstention, restricted recommendations, or deferral when support is insufficient. |
| **Sensitivity analysis.** Stability of recommendations and value under plausible alternative specifications of omitted interventions, interactions, or background policies. |
| **Benchmark alignment.** Dataset records co-interventions and task reflects joint actionability. |
| **Deployment drift.** State how changes in the actionable set will be monitored. |

## 7. Conclusion

We argue that personalised treatment decision support at deployment requires holistic intervention modelling, as the prevailing practice of optimising ITE-based objectives over restricted intervention subsets can induce OIB through intervention space misspecification. We show that this can lead to misaligned recommendations and worse achievable outcomes. We discuss alternative views and clarify when restricted-intervention analyses are appropriate, while emphasising that claims of personalised decision support should be scoped to the actionable intervention set at deployment. These concerns are particularly salient for clinically complex and vulnerable populations with comorbidities, where treatment–treatment interactions are common and misaligned recommendations can exacerbate inequities, reinforcing the need for principled, deployment-aligned causal approaches. Finally, we provide recommendations to align causal AI with personalised treatment objectives.

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

# A. Proof (Potential outcomes framework)

Here, we provide proof of Theorem 4.1 using the Potential Outcomes framework, which is restated below.

**Theorem A.1** (Omitted intervention bias). *For a personalised treatment problem with multiple actionable (composite) interventions, if decisions are supported by an ITE estimand that models a strict subset of interventions, then omitted intervention bias can arise, even when causal effects for the included interventions are correctly identified.*

*Proof.* We work in the potential outcomes framework with two actionable interventions $T_1, T_2 \in \{0, 1\}$ and joint potential outcomes $\{Y(t_1, t_2) : t_1, t_2 \in \{0, 1\}\}$ (Rubin, 2005). We index an individual by common pre-intervention covariates $\mathbf{X} = \mathbf{x}$. For identification from observational data, we allow additional pre-intervention covariates $\mathbf{X}_1 = \mathbf{x}_1$ and $\mathbf{X}_2 = \mathbf{x}_2$ (for example, intervention-specific confounders), all measured prior to $(T_1, T_2)$.

We assume the standard conditions for identification from observational data: (i) *Consistency*: if $(T_1, T_2) = (t_1, t_2)$ then $Y = Y(t_1, t_2)$; (ii) *Conditional exchangeability*: $\{Y(t_1, t_2)\} \perp (T_1, T_2) \mid \mathbf{X}, \mathbf{X}_1, \mathbf{X}_2$ for all $(t_1, t_2)$; (iii) *Positivity*: $\mathbb{P}(T_1 = t_1, T_2 = t_2 \mid \mathbf{X} = \mathbf{x}, \mathbf{X}_1 = \mathbf{x}_1, \mathbf{X}_2 = \mathbf{x}_2) > 0$ for all $(t_1, t_2)$ and all $(\mathbf{x}, \mathbf{x}_1, \mathbf{x}_2)$ in the support of $(\mathbf{X}, \mathbf{X}_1, \mathbf{X}_2)$ (Hernan & Robins, 2025; Imbens & Rubin, 2015).

● *Scenario A*. *Holistic intervention modelling.* Define the joint-intervention value for an individual with $(\mathbf{X}, \mathbf{X}_1, \mathbf{X}_2) = (\mathbf{x}, \mathbf{x}_1, \mathbf{x}_2)$:

$$V(t_1, t_2; \mathbf{x}, \mathbf{x}_1, \mathbf{x}_2) := \mathbb{E}[Y(t_1, t_2) \mid \mathbf{X} = \mathbf{x}, \mathbf{X}_1 = \mathbf{x}_1, \mathbf{X}_2 = \mathbf{x}_2]. \quad (11)$$

By conditional exchangeability and then consistency, and using positivity to ensure the conditioning event has non-zero probability, we obtain the identified form:

$$\begin{aligned} V(t_1, t_2; \mathbf{x}, \mathbf{x}_1, \mathbf{x}_2) &= \mathbb{E}[Y(t_1, t_2) \mid T_1 = t_1, T_2 = t_2, \mathbf{X} = \mathbf{x}, \mathbf{X}_1 = \mathbf{x}_1, \mathbf{X}_2 = \mathbf{x}_2], \\ &= \mathbb{E}[Y \mid T_1 = t_1, T_2 = t_2, \mathbf{X} = \mathbf{x}, \mathbf{X}_1 = \mathbf{x}_1, \mathbf{X}_2 = \mathbf{x}_2]. \end{aligned} \quad (12)$$

The holistic personalised treatment objective is

$$(t_1^\star, t_2^\star) \in \arg\max_{t_1, t_2 \in \{0, 1\}} V(t_1, t_2; \mathbf{x}, \mathbf{x}_1, \mathbf{x}_2). \quad (13)$$

Next, we derive the corresponding objectives under restricted intervention modelling, considering two common practices: **(B1)** the omitted intervention is ignored as background and not optimised over, and **(B2)** the omitted intervention is treated as a covariate and then treated as fixed when optimising. Both scenarios need the above three assumptions for identifiability only for the intervention considered for modelling.

● *Scenario B1*. *Restricted intervention modelling when the omitted intervention is ignored as background.* A common restricted practice optimises only over $T_1$ using the conditional mean outcome given $T_1$:

$$V(t_1; \mathbf{x}, \mathbf{x}_1, \mathbf{x}_2) := \mathbb{E}[Y(t_1) \mid \mathbf{X} = \mathbf{x}, \mathbf{X}_1 = \mathbf{x}_1, \mathbf{X}_2 = \mathbf{x}_2]. \quad (14)$$

Again, by conditional exchangeability and then consistency, and using positivity w.r.t. $T_1$ to ensure the conditioning event has non-zero probability, we obtain the identified form:

$$\begin{aligned} V(t_1; \mathbf{x}, \mathbf{x}_1, \mathbf{x}_2) &= \mathbb{E}[Y(t_1) \mid T_1 = t_1, \mathbf{X} = \mathbf{x}, \mathbf{X}_1 = \mathbf{x}_1, \mathbf{X}_2 = \mathbf{x}_2], \\ &= \mathbb{E}[Y \mid T_1 = t_1, \mathbf{X} = \mathbf{x}, \mathbf{X}_1 = \mathbf{x}_1, \mathbf{X}_2 = \mathbf{x}_2]. \end{aligned} \quad (15)$$

It can be written as expectation over $T_2$,

$$V(t_1; \mathbf{x}, \mathbf{x}_1, \mathbf{x}_2) = \sum_{t_2 \in \{0,1\}} \mathbb{E}[Y \mid T_1 = t_1, T_2 = t_2, \mathbf{X} = \mathbf{x}, \mathbf{X}_1 = \mathbf{x}_1, \mathbf{X}_2 = \mathbf{x}_2] \, \mathbb{P}(T_2 = t_2 \mid T_1 = t_1, \mathbf{X} = \mathbf{x}, \mathbf{X}_1 = \mathbf{x}_1, \mathbf{X}_2 = \mathbf{x}_2).$$

$$(16)$$

Using Eq. (12), the conditional expectation term equals the identified holistic value at $(t_1, t_2)$, so

$$V(t_1; \mathbf{x}, \mathbf{x}_1, \mathbf{x}_2) = \sum_{t_2 \in \{0,1\}} V(t_1, t_2; \mathbf{x}, \mathbf{x}_1, \mathbf{x}_2) \, \mathbb{P}(T_2 = t_2 \mid T_1 = t_1, \mathbf{X} = \mathbf{x}, \mathbf{X}_1 = \mathbf{x}_1, \mathbf{X}_2 = \mathbf{x}_2). \quad (17)$$

Thus, the restricted objective optimises a *mixture* of joint-intervention values, with weights given by the observed conditional distribution of the omitted co-intervention among those with $T_1 = t_1$. The corresponding restricted objective is

$$\max_{t_1 \in \{0,1\}} V(t_1; \mathbf{x}, \mathbf{x}_1, \mathbf{x}_2). \tag{18}$$

● *Scenario B2. Restricted intervention modelling when the omitted intervention is conditioned on (treated as fixed context).* Another common practice conditions on the co-intervention level and then optimises over $T_1$:

$$V(t_1; t_2, \mathbf{x}, \mathbf{x}_1, \mathbf{x}_2) := \mathbb{E}[Y(t_1) \mid T_2 = t_2, \mathbf{X} = \mathbf{x}, \mathbf{X}_1 = \mathbf{x}_1, \mathbf{X}_2 = \mathbf{x}_2]. \tag{19}$$

Again, by conditional exchangeability and then consistency, and using positivity w.r.t. $T_1$ to ensure the conditioning event has non-zero probability, we obtain the identified form:

$$\begin{aligned} V(t_1; t_2, \mathbf{x}, \mathbf{x}_1, \mathbf{x}_2) &= \mathbb{E}[Y(t_1) \mid T_1 = t_1, T_2 = t_2, \mathbf{X} = \mathbf{x}, \mathbf{X}_1 = \mathbf{x}_1, \mathbf{X}_2 = \mathbf{x}_2], \\ &= \mathbb{E}[Y \mid T_1 = t_1, T_2 = t_2, \mathbf{X} = \mathbf{x}, \mathbf{X}_1 = \mathbf{x}_1, \mathbf{X}_2 = \mathbf{x}_2]. \end{aligned} \tag{20}$$

By Eq. (12), we have the identity
$$V(t_1; t_2, \mathbf{x}, \mathbf{x}_1, \mathbf{x}_2) = V(t_1, t_2; \mathbf{x}, \mathbf{x}_1, \mathbf{x}_2), \tag{21}$$

but the *optimisation target* is different because $t_2$ is treated as fixed:

$$\max_{t_1 \in \{0,1\}} V(t_1; t_2, \mathbf{x}, \mathbf{x}_1, \mathbf{x}_2). \tag{22}$$

**Value gap (key idea).** The holistic objective (13) optimises over $(t_1, t_2)$, whereas the restricted objectives (18) and (22) optimise over $t_1$ only, with $T_2$ either averaged under a background distribution (17) or fixed at a conditioned level (22). Therefore, for any $(\mathbf{x}, \mathbf{x}_1, \mathbf{x}_2)$,

$$\max_{t_1, t_2} V(t_1, t_2; \mathbf{x}, \mathbf{x}_1, \mathbf{x}_2) \ \geq\ \max_{t_1} V(t_1; \mathbf{x}, \mathbf{x}_1, \mathbf{x}_2), \qquad \max_{t_1, t_2} V(t_1, t_2; \mathbf{x}, \mathbf{x}_1, \mathbf{x}_2) \ \geq\ \max_{t_1} V(t_1; t_2, \mathbf{x}, \mathbf{x}_1, \mathbf{x}_2), \tag{23}$$

since both restricted procedures optimise over a smaller set of intervention choices than the holistic objective. Moreover, the inequalities in (23) can be *strict* when the joint response surface is not separable (for example, when there are interactions) and when the background distribution in (17) or the fixed level in (22) does not coincide with the jointly optimal choice of $t_2$ for the individual. This yields a strictly positive value gap, which is OIB in value, as instantiated by the same constructions used in the main text. $\qquad\square$

## B. Example for Scenario B2

*Example* B.1 (Scenario B2). Use the same individual and parameters as in Example 4.3, so $g(\mathbf{x}) = 1$ and $g_1(\mathbf{x}_1) = g_2(\mathbf{x}_2) = 0$ with
$$\alpha_1 = -0.2, \quad \alpha_2 = 0, \quad \beta_0 = 2.5, \quad \beta_1 = \beta_2 = 0.$$

In Scenario B2, conditioning on $t_2 = 0$ yields from Eq. (8)

$$V(t_1; t_2 = 0, \mathbf{x}, \mathbf{x}_1, \mathbf{x}_2) = \alpha_1 t_1,$$

so the restricted objective selects $t_1^\dagger = 0$ and achieves $\max_{t_1} V(t_1; t_2 = 0, \cdot) = 0$. However, the holistic optimum from Example 4.3 achieves $\max_{t_1, t_2} V(t_1, t_2; \cdot) = 2.3$. Therefore, the strict value gap is again $2.3 - 0 = 2.3$.

## C. Simulation Results

We provide simulation to study the phenomenon in Section 4 under a standard outcome-regression (S-learner) (Künzel et al., 2019) pipeline. In the simulation, decisions are made using model-predicted conditional means, as in practice. For evaluation, we use the data-generating process (DGP) implied conditional mean (an oracle value) to measure policy value without conflating it with outcome noise. This oracle is used for evaluation only, not for decision-making.

Estimated OIB (value gap) vs interaction strength (model A)

Estimated OIB (value gap) vs interaction strength (model B)

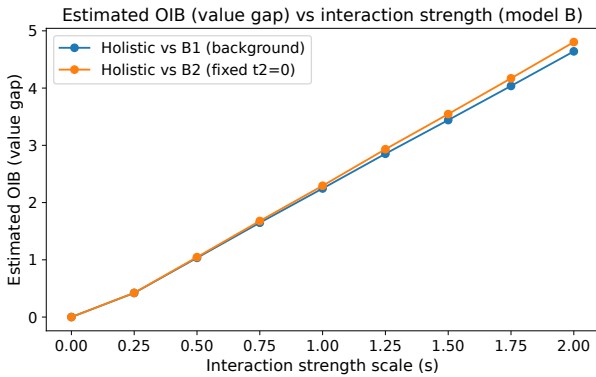

*(a)* Estimated OIB (value gap) with Model A (linear S-learner with engineered interaction features).

*(b)* Estimated OIB (value gap) with Model B (gradient boosting S-learner on raw inputs).

*Figure 3.* Estimated OIB (value gap) versus interaction strength $s$.

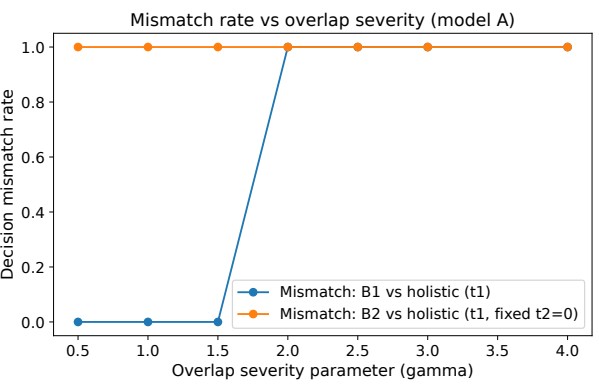

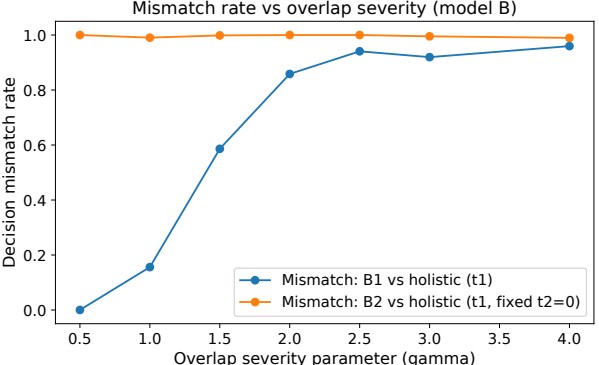

*(a)* Decision-mismatch rate with Model A (linear S-learner with engineered interaction features).

*(b)* Decision-mismatch rate with Model B (gradient-boosting S-learner on raw inputs).

*Figure 4.* Decision-mismatch rate versus overlap severity $\gamma$.

**DGP.** We simulate a single covariate $X \sim \mathcal{N}(0,1)$, two binary jointly actionable interventions $T_1, T_2 \in \{0,1\}$, and an outcome

$$Y = \alpha_1 T_1 + \alpha_2 T_2 + \beta(X) T_1 T_2 + \varepsilon, \qquad \beta(X) = s(b_0 + b_1 X), \qquad \varepsilon \sim \mathcal{N}(0, \sigma_\varepsilon^2),$$

with fixed coefficients $(\alpha_1, \alpha_2) = (-0.2, 0)$ and $(b_0, b_1) = (2.5, 0.5)$. The interaction scale $s \geq 0$ controls the strength of non-additivity. Treatment assignment follows logistic mechanisms with overlap severity parameter $\gamma > 0$:

$$T_1 \sim \text{Bernoulli}(\sigma(\gamma \cdot 0.6X)), \qquad T_2 \sim \text{Bernoulli}(\sigma(\gamma \cdot (-2.0 + 0.6X + 0.2T_1))),$$

where $\sigma(z) = 1/(1 + e^{-z})$. Increasing $\gamma$ yields more extreme propensities and hence poorer overlap, and the negative intercept makes $T_2 = 1$ comparatively rare, stressing positivity for some joint assignments.

**Modelling choices.** (i) We fit a *Holistic:* S-learner outcome model $\hat{m}(x, t_1, t_2) \approx \mathbb{E}[Y \mid X = x, T_1 = t_1, T_2 = t_2]$ and choose $(\hat{t}_1^{\text{hol}}(x), \hat{t}_2^{\text{hol}}(x)) \in \arg\max_{(t_1, t_2) \in \{0,1\}^2} \hat{m}(x, t_1, t_2)$; (ii) *B1 (ignore as background):* fit a restricted S-learner $\hat{m}_{\text{B1}}(x, t_1) \approx \mathbb{E}[Y \mid X = x, T_1 = t_1]$ and choose $\hat{t}_1^{\text{B1}}(x) \in \arg\max_{t_1 \in \{0,1\}} \hat{m}_{\text{B1}}(x, t_1)$; (iii) *B2 (fixed context):* use the same $\hat{m}(x, t_1, t_2)$ as holistic but fix $t_2$ to a context value (here $t_2 = 0$) and choose $\hat{t}_1^{\text{B2}}(x) \in \arg\max_{t_1 \in \{0,1\}} \hat{m}(x, t_1, t_2 = 0)$. We report results under two modelling choices for holistic case: (A) a linear S-learner with engineered interaction features to minimise estimation error and isolate action-space restriction, and (B) a flexible regressor (gradient boosting) trained on raw inputs $(X, T_1, T_2)$ only, to mimic a more typical machine-learning workflow without manual feature engineering.

**Evaluation metrics.** We evaluate policies on a held-out test set using the DGP-implied conditional mean (oracle value)

$$V(t_1, t_2; x) = \mathbb{E}[Y(t_1, t_2) \mid X = x] = \alpha_1 t_1 + \alpha_2 t_2 + \beta(x) t_1 t_2.$$

For Scenario B1, we evaluate under the true background assignment mechanism for the omitted action,

$$V(t_1; x) = \mathbb{E}_{T_2 \sim p(\cdot \mid x, t_1)}[V(t_1, T_2; x)],$$

where $p(\cdot \mid x, t_1)$ is induced by the DGP assignment for $T_2$. We report (i) the estimated OIB (value gap) $\mathbb{E}[V(\hat{t}_1^{\text{hol}}, \hat{t}_2^{\text{hol}}; X) - V(\hat{t}_1^{\text{B1}}; X)]$ and $\mathbb{E}[V(\hat{t}_1^{\text{hol}}, \hat{t}_2^{\text{hol}}; X) - V(\hat{t}_1^{\text{B2}}; t_2 = 0, X)]$, and (ii) the mismatch rates $\mathbb{P}(\hat{t}_1^{\text{hol}} \neq \hat{t}_1^{\text{B1}})$ and $\mathbb{P}(\hat{t}_1^{\text{hol}} \neq \hat{t}_1^{\text{B2}})$.

**Results.** Figures 3–4 provide an empirical demonstration of OIB under two S-learner instantiations. First, Figure 3 shows that the estimated OIB (value gap) between the holistic joint-optimisation policy and the restricted policies increases monotonically with the interaction scale $s$. When $s = 0$ (approximately additive response surface), the gap is near zero; as $s$ increases, non-additivity strengthens and the cost of restricting the actionable set grows, yielding substantial value loss under both B1 (background) and B2 (fixed-context) restrictions. This aligns with the central mechanism in Section 4: when outcomes depend on coordinated combinations (here through an interaction term), optimising only a subset of actions can be systematically suboptimal even if the restricted policy is internally consistent with its estimand.

Second, Figure 4 shows that action-space restriction can induce pronounced decision mismatch in the optimised component $T_1$, and that this mismatch increases as overlap deteriorates (larger $\gamma$). For B1, mismatch is low under benign overlap but rises sharply as the omitted intervention becomes rarer and the induced background distribution for $T_2$ changes across $T_1$ levels. Intuitively, the restricted B1 decision rule learns a mixture objective whose effective weights depend on the realised co-intervention mechanism; as overlap worsens, these induced weights become more extreme, amplifying divergence from the joint optimum. For B2, mismatch remains close to one across the overlap range because $T_2$ is fixed at $t_2 = 0$ during optimisation, while the holistic policy frequently selects $(t_1, t_2) = (1, 1)$ when interactions are beneficial. Thus, B2 differs from the holistic policy purely because it solves a smaller optimisation problem, not because it uses a different outcome model.

Finally, the similarity of the value-gap curves for Models A and B indicates that the phenomenon is not an artefact of a particular learner: it appears both when the outcome model is near-correctly specified (Model A with engineered interactions) and when a more typical flexible learner is trained on raw inputs (Model B). **Overall, these results are consistent with our formal argument: restricting the intervention set can lead to both (i) a strict value gap and (ii) decision mismatch, and the effects are exacerbated when interactions are strong and overlap is poor.**

## D. Related Work

This section situates our position relative to existing work on (i) ITEs, (ii) ITE benchmarks, software pipelines, and evaluation conventions, (iii) applied personalisation practice, (iv) emerging methods for ITE estimation under multiple simultaneous interventions, and (v) adjacent notions to OIB. Since Section 2 already summarises the prevalence of restricted intervention modelling, we focus here on organising the literature by modelling targets and evaluation conventions, and on clarifying how these strands relate to deployment-time personalised decision support.

### D.1. Individualised treatment effects

Individualised treatment effects (ITEs), also referred to as conditional average treatment effects (CATEs) or heterogeneous treatment effects (HTEs), arise naturally from the potential outcomes framework as conditional contrasts of counterfactual outcomes given covariates (Rubin, 2005; Imbens & Rubin, 2015; Hernan & Robins, 2025). In practice, much of the methodological literature targets the setting with a *single* treatment variable as the optimised decision lever (binary, categorical, or continuous), and develops estimators for conditional effects under that formulation. A prominent family is *meta-learners*, which reduce ITE estimation to modal-agnostic supervised learning subproblems, including S-, T-, X-, R-, and doubly robust style learners and related orthogonalisation ideas (Künzel et al., 2019; Curth & van der Schaar, 2021b; Kennedy, 2023; Melnychuk et al., 2025a). Another major line uses *tree and forest based* estimators that partition covariate space to capture heterogeneity and enable policy-style use through effect-guided splits (Athey & Imbens, 2016; Wager & Athey, 2018). More recent work integrates representation learning and deep models for ITE estimation, typically still under a single-intervention interface, motivated by high-dimensional covariates and flexible function approximation (Johansson et al., 2016; Shalit et al., 2017; Curth & van der Schaar, 2021a; Melnychuk et al., 2022; Curth et al., 2024). Across these families, the prevailing modelling and evaluation conventions typically treat one intervention variable as the decision lever, which is precisely the default that our paper argues can be misaligned with deployment-time personalised treatment when multiple interventions are jointly actionable.

## D.2. Benchmarks, software pipelines, and evaluation conventions for ITE

Widely used benchmarks for ITE estimation frequently instantiate a *single* treatment variable (semi-synthetic or observational), such as Jobs, IHDP, Twins, and ACIC-style tasks (Hill, 2011; Shalit et al., 2017; Dorie et al., 2019). These benchmarks have been valuable for methodological comparison, but their task structure does not reflect joint actionability at deployment. More broadly, recent work has stressed that benchmark choices can encode implicit assumptions and mask failure modes that matter for deployment claims (Curth et al., 2021). Consistent with this, common evaluation protocols emphasise effect-estimation error metrics (for example PEHE and influence functions) and ranking metrics in uplift settings, while decision-focused evaluation (policy value, regret, value loss) is less consistently reported outside policy-learning oriented work (Hill, 2011; Radcliffe & Surry, 2011; Kitagawa & Tetenov, 2018; Alaa & Van Der Schaar, 2019; Athey & Wager, 2021). Software packages for effect estimation also reinforce the single-intervention interface by design: widely used software packages and pipelines, such as DoWhy, EconML and CausalML, provide APIs, estimands, and examples that centre on one treatment variable and its ITE, even when applications acknowledge multiple concurrent actions (Oprescu et al., 2019; Chen et al., 2020; Sharma & Kiciman, 2020). Survey evidence aligns with this pattern: Yao et al. (2021)'s benchmark compilation contains no benchmark designed for jointly actionable multi-intervention vectors, and Cheng et al. (2022)'s comparison of widely used causal inference tools does not identify any package with first-class support for modelling or optimising multi-intervention vectors. Our recommendations in Section 6 are intended to make these interfaces and benchmark tasks more explicit about actionability and decision scope. This "single-treatment-by-default" interface is a pragmatic choice for usability and benchmarking, but it can encourage broad personalisation claims that exceed what the benchmarked estimand and evaluation actually support.

## D.3. Applied ITE research

In applied work, personalisation is often operationalised by selecting a focal intervention and estimating heterogeneity for that intervention, while other concurrent interventions are either adjusted for as covariates or left outside the intervention set entirely. In healthcare, examples include heterogeneous effect modelling for specific drug decisions (for example anticoagulation choices in atrial fibrillation) (Ngufor et al., 2023; Xu et al., 2023), as well as counterfactual prediction and regime-style modelling on ICU datasets where the modelling target is often a restricted intervention component even when care is intrinsically multi-interventional (Xiong et al., 2024). In economics and public policy, prominent settings estimate heterogeneous impacts of a focal programme (for example youth employment interventions) while other simultaneous policy actions remain in the background (Davis & Heller, 2017). In marketing, uplift modelling commonly targets the incremental effect of a single campaign versus control at the individual level, despite multiple available actions (Radcliffe & Surry, 2011; Goldenberg et al., 2025). These formulations are often scientifically coherent under retrospective or incremental objectives, but they highlight a recurring pattern: the estimand and evaluation privilege single-intervention effects even when deployment-time decisions can involve jointly selecting multiple actionable, interacting interventions, which motivates the intervention-set clarity and value-based evaluation emphasis in this paper.

## D.4. Emerging methodology for composite and multi-intervention personalised decision support

A smaller, emerging body of work aims to move beyond single-intervention formulations towards settings with multiple simultaneous interventions, bundle-style actions, multi-arm choices, and high-dimensional treatment vectors, motivated by realistic decision-making where actions are not isolated. This includes work on multi-cause settings and high-dimensional or bundled treatments (Wang & Blei, 2019; Zou et al., 2020), as well as early causal machine learning efforts that treat multiple treatments explicitly but often under restrictive assumptions (for example binary-only bundles, limited action types, or modelling choices that scale poorly as the joint action space expands) (Qian et al., 2021; Schweisthal et al., 2023; Chauhan et al., 2025a). In adjacent communities, multi-treatment uplift and multi-action targeting settings also formalise the challenges of choosing among multiple alternatives and scaling decision rules beyond a single binary lever (Zhao et al., 2017; Wei et al., 2024). Overall, the emerging literature substantiates both sides of the practical tension: it confirms the importance of multi-intervention decision targets, while also illustrating why data scarcity, overlap, and combinatorial action growth make holistic optimisation difficult in observational settings. This motivates the paper's stance that feasibility constraints do not remove the *conceptual* need to align claims, estimands, and evaluation with the actionable intervention set at deployment.

## D.5. Omitted intervention bias and adjacent literatures

The term *omitted intervention bias* is not standard in the causal inference, but closely related phenomena have been studied under several well-established framings. First, causal inference has long emphasised that causal effects require *well-defined interventions*. When what is modelled as a single treatment variable is in fact a *compound* or *multi-component* intervention, different "versions" of treatment may be implicitly pooled within the same treatment level. In that case, the estimated contrast can correspond to a mixture over versions whose composition may vary across covariates, complicating causal interpretation and potentially shifting the implied estimand away from the decision problem of interest (Hernán & VanderWeele, 2011; VanderWeele & Hernan, 2013). This literature clarifies that the difficulty is not merely statistical, but conceptual: coarse treatment definitions can encode implicit background policies over omitted or unmodelled intervention components (Hernán & Taubman, 2008).

Second, in clinical trials and observational study appraisal, *co-interventions* (additional interventions received alongside the focal intervention) are explicitly recognised as a pathway by which effect estimates may be distorted if co-interventions differ systematically across groups or are handled inconsistently. Risk-of-bias frameworks for non-randomised studies, such as ROBINS-I, operationalise this concern by requiring reviewers to assess whether deviations from intended interventions and related co-interventions could have affected outcomes and comparability (Sterne et al., 2016). This literature is typically framed around bias in effect estimation for a pre-specified intervention contrast, whereas our OIB framing targets the deployment-time *decision objective* when multiple interventions are actionable and should be jointly optimised.

Third, econometrics has discussed closely aligned issues under the name *omitted treatment bias* in settings with *multi-dimensional treatments*. A canonical example arises in judge-IV designs for sentencing, where an instrument shifts multiple components of punishment (e.g., incarceration, sentence length, supervision conditions) but empirical analyses often focus on a single focal component; omitting other shifted treatment dimensions can bias the estimated effect of the focal treatment (Mueller-Smith, 2015). Recent work on *learned treatment representations* for high-dimensional treatments in IV settings similarly highlights that dimension reduction or partial modelling can omit relevant treatment variation, inducing bias relative to downstream causal targets (Lin et al., 2025). This strand is particularly close in spirit to our setting, but focuses on identification and estimation of causal parameters under multi-dimensional treatments, rather than the value gap induced by restricted optimisation in personalised decision support.

Finally, OIB is related to general *omitted variable bias* (OVB) and sensitivity-analysis frameworks in causal inference and causal machine learning, which primarily study bias from *unobserved confounders* or other omitted variables that are associated with both the focal intervention and the outcome after conditioning (Chernozhukov et al., 2022; Frauen et al., 2024). While these frameworks can also be applied when the omitted variable is another *actionable co-intervention* that is correlated with the modelled intervention, our contribution differs in emphasis: we focus on how restricting the *optimised intervention set* changes the personalised decision target and can reduce achievable outcomes, complementing OVB work that quantifies parameter bias for a fixed estimand.

In contrast to these adjacent strands, our paper isolates a deployment-time *decision-target mismatch*: even if the causal effect for the included intervention(s) is well-defined and correctly identified for the restricted estimand under standard assumptions, optimising over a strict subset of actionable interventions can still yield misaligned personalised recommendations and a strict value gap relative to holistic joint optimisation.

