# OpenReview forum: "Position: Causal AI for Personalised Treatments Needs Holistic Approach"
_ICML.cc/2026/Position_Paper_Track — Submitted to ICML 2026 Position Paper Track_

### Official Review · Reviewer_j12a · 2026-03-10

**Significance:** 2
**Argument Clarity:** 2
**Rating:** 2
**Confidence:** 5

**Questions:**

See the weaknesses

**Alternative Views Section:**

Yes

**Compliance With Llm Reviewing Policy A Conservative:**

Affirmed.

**Discussion Potential:**

2

**Final Justification:**

While the stated position is important, but the position itself is obvious or well known in the literature. Hence, it may not bring up any new discussions in the community. Rebuttals also highlight the same point that OIB is common and should be considered. But this is already known and does not provide new insights or observations.

**Paper Summary:**

This paper argues against the practice of restricted set of interventions in individualized treatment effects when actual required intervention set is bigger and actionable. The paper formalizes the idea of omitted intervention bias and how omitted intervention bias occurs if the intervention set is a subset of actual intervention set. The paper then provides alternative views and best practices for reducing the gap between desired objective of individual treatment effect estimation and the objectives considered in practice,

**Position:**

Yes

**Position In Title:**

Yes

**Related Work:**

3

**Strengths And Weaknesses:**

The paper is well written and easy to understand. The paper has the following weaknesses.

1. If the actual intervention set is different from the desired intervention set, the estimated individual causal effect will be biased. This fact is well known and the reason for using a subset of interventions is due to fundamental limitations rather than technical limitations or oversight/ignorance. For example, if the causal effect is estimated using observational data, having a high-dimensional treatment often suffers from violations of the positivity assumption. Such data scarcity cannot be solved easily in real-world settings. On the other hand, if the causal effect estimation is performed using randomized experiments, those experiments have to be performed ethically and they are time-consuming. The data scarcity issue has been acknowledged in Section 5.5 but the paper does not suggest any useful strategies to address this issue.

2. The paper uses the phrase: "even when causal effects are correctly identified for the included interventions" several times. However, it does not provide any useful insights in the context of the paper's idea. When the intervention set is not the correct one, obtaining correct or incorrect causal effects with respect to the included interventions does not have any significance as the estimation is biased.

3. The results of Section 4.2 do not provide any useful insights beyond what is already known in the literature.

**Support:**

3

---

> ### Author Rebuttal · Authors · 2026-03-27
>
> We are grateful for the reviewer’s time and thoughtful review, and for noting that the paper is well written and easy to understand.
>
> ## W1. Estimation bias and feasibility
> We would like to clarify an important distinction. Our paper is **not** about ordinary estimation bias in the treatment effect of the included interventions. Rather, it studies a decision-target mismatch: when multiple interventions are actionable at deployment, but the estimand and optimisation are defined over only a strict subset, the resulting decision rule can be misaligned with the deployment-time personalised treatment objective and yield worse achievable outcomes. In our formal analysis, the relevant causal effects for both the holistic and restricted problems are assumed to be correctly identified; the issue is that they correspond to different optimisation problems.
>
> We fully agree that data scarcity, positivity violations, and experimental constraints are major reasons why restricted intervention modelling is common in practice. However, these constraints do not remove the conceptual mismatch or its potential harmful consequences for personalised decision support. This is precisely why our position is needed: to encourage appropriate reporting (refer to 'Recommended Reporting Template' box) that avoids such consequences, and to encourage research that addresses these constraints.
>
> Regarding strategies, we do point to several research directions in Section 6.7, including structured action representations, meta-learning and information-sharing across related intervention combinations, overlap-aware objectives (e.g. [1]), abstention/deferral, and sensitivity analysis. Of course, these are not complete solutions, but they are intended as concrete directions consistent with the paper’s role as a position paper.
>
> ## W2. Why emphasise “*even when causal effects are correctly identified for the included interventions*”?
> This phrase matters because it shows the problem is not that the restricted analysis is badly estimated, but that it solves the wrong optimisation problem. Its purpose is to isolate OIB from standard concerns such as confounding, model misspecification, or ordinary estimation error. The point is that even if the restricted intervention-set effect is correctly identified for that restricted problem, it may still be insufficient for deployment-time personalised objective when additional actionable interventions are omitted. In our framing, OIB is therefore not an estimation bias; it is a value gap induced by solving a restricted optimisation problem instead of the holistic one.
>
> ## W3. Section 4.2 does not provide useful insight beyond existing literature
> We respectfully disagree. We agree that the core intuition may appear straightforward once stated, but we believe the constructive result is still useful because it (1) formalises the issue in the language of CATE/ITE-based personalised decision support, (2) distinguishes it from ordinary estimation bias, and (3) demonstrates that decision mismatch and a strict value gap can arise under two common restricted-modelling scenarios. Additionally, while the result may seem to establish an obvious point, it is crucial for convincing readers, through a formal mathematical argument, that OIB can arise even when the restricted estimand is correctly identified, which, to the best of our knowledge, has not previously been formalised in the CATE/ITE personalised decision-support literature.
>
>
> [1] Melnychuk, et al (2025). Overlap-adaptive regularization for conditional average treatment effect estimation. arXiv:2509.24962.

---

> > ### Author Rebuttal · Reviewer_j12a · 2026-04-01
> >
> > **W1.**  I understand that the paper studies decision-target mismatch. I believe that practitioners who rely on an estimand defined over a strict subset of actionable interventions often do so due to practical constraints or limitations. While accurate reporting helps make these differences explicit and acknowledges the limitations of using a restricted subset of interventions, the core idea itself feels relatively well understood, and it is generally recognized that decision-target mismatch can lead to issues.
> >
> > **W3.** I feel that some of these statements may not require extensive formalization or justification. My sense is that many causality researchers and practitioners are already aware of omitted intervention bias, so dedicating nearly an entire section to this may be more than necessary.

---

### Official Review · Reviewer_xVAa · 2026-03-13

**Significance:** 3
**Argument Clarity:** 3
**Rating:** 5
**Confidence:** 4

**Questions:**

- Are you aware of benchmark-level problems where the OIB can be estimated?

**Alternative Views Section:**

Yes

**Compliance With Llm Reviewing Policy A Conservative:**

Affirmed.

**Discussion Potential:**

3

**Final Justification:**

The authors have addressed my remaining concerns regarding a benchmark-level problem to estimate OIB on. I maintain the paper is well-written, informative, and is likely to generate discussion regarding the application of personalized treatment effect estimates.

**Paper Summary:**

This paper argues that estimates of the individualized treatment effect (ITE; a.k.a. conditional average treatment effects, CATE) are marred in practice by focusing on a subset of potential interventions. That is, if some set of K treatments, $T_1, \dots, T_K$, are available, existing methods only effectively model the effects of some subset, $T_1, \dots, T_S$, with $S < K$. In the corresponding decision problem, where one is to choose which interventions to apply, suboptimal decisions occur. This is both theoretically relevant, and practically relevant, as multi-treatment regimes may be the standard rather than the norm in some applications. This is formalized in a constructive example which shows that the gap between the value of actions under the optimal, fully-modeled multi-intervention model w.r.t. the sub-optimal model can be strictly positive. A number of alternative views are discussed, including data scarcity and single-treatment applications.

**Position:**

Yes

**Position In Title:**

Yes

**Related Work:**

3

**Strengths And Weaknesses:**

### Strengths

- The position is articulated well, and the paper is overall easy-to-read. The motivation is concrete from the application perspective, this is connected to a gap in the literature, and a simple but effective analysis is undertaken. The manuscript does a very good job of being accessible in its arguments, which is imperative for a position paper.
- The alternative views sections is extremely fair in identifying competing views. In particular, alternative views are explained in a way that doesn't diminish, but rather contextualizes, them. For example, the alternative view that "interactions are negligible" is acknowledged as credible in some cases, but a strong assumption (additivity of treatment effects) that should be explicitly stated, and only realistic in certain applications.
- While I have some objections to the main "theorem," it serves as an intuitive constructive example of when the non-holistic approach may go awry.
- There is a concrete call to action, that I believe can catalyze conversations within the community. In particular, the work highlights a specific gap in both methodology and stated assumptions in practice, and can help draw attention to this issue.


### Weaknesses

- I think the main "theorem" is somewhat trivial; it's somewhat obvious that optimizing a function over a subset of its arguments can obtain suboptimal results. This is better stated as an example, in my opinion.
  - Indeed, the proof is incomplete without the explicit construction in Example 4.3.
- This is minor, but given that the venue is ICML (and therefore the audience mostly machine learning researchers), it seems advantageous to refer to the CATE instead of the ITE (the former being the standard term in ML). The result is also presented in both Pearlian and potential outcomes formalisms (the former in the main manuscript, the latter in an appendix), which again makes the work more approachable to a larger audience.
- While there are several conceptual examples given throughout the text, and the constructive example is helpful, it would be useful to identify an example where OIB can be illustrated that is more of a benchmark than toy problem.
  - I understand part of the argument is that this is difficult because benchmarks rarely model sophisticated interactions (c.f. Section 6.4), but this would still increase the impact of this work considerably.

### Editorial Remarks

- I don't think the introduction _quite_ follows the guidelines that the position itself be bolded (it instead just bolds "in this paper, we take a position that").
- The use of text boxes with text smaller than the generic font size may violate the general style guide.
- I think the title is missing an article (i.e., it should be "Causal AI for Personalised Treatments Needs **a** Holistic Approach", or "Needs **the** Holistic Approach).

**Support:**

3

---

> ### Author Rebuttal · Authors · 2026-03-27
>
> We are grateful for the reviewer’s time and thoughtful review, for the positive assessment of the paper’s accessibility and clarity, fair treatment of alternative views, concrete call to action, and potential to catalyse discussion in the community, and for the helpful suggestions. We respond to all comments below; we add all items labeled with **Action** in the camera-ready version.
>
> ## W1. The theorem is somewhat trivial
>
> Thank you for the comment. We would like to clarify the purpose of the theoretical result. It is _not_ intended to be technically complex, but rather to provide a precise and rigorous formalisation of the mismatch we identify. In particular, it isolates the phenomenon from standard estimation issues and shows that it can arise even when the relevant causal effects are correctly identified.
>
>  We are happy to revise the wording in that direction. At the same time, we think the formal result remains useful because it shows something precise that is easy to blur in discussion: a restricted intervention-set formulation can induce OIB and decision mismatch even when the causal effects for the included interventions are correctly identified. In the paper, we make this concrete under two common restricted-modelling scenarios, B1 (ignores omitted interventions from modelling) and B2 (treats omitted interventions as covariates), and under both Potential Outcome and Structural Causal Model frameworks. The explicit constructions are given in Example 4.3 and Appendix B.1.
>
> **Action:** We will tone down our theorem and use it as auxiliary tool to motivate our arguments.
>
> ## W2. Use CATE terminology and improve accessibility to the ML audience
> Thank you for this suggestion. **Action**: We will revise the terminology in the camera-ready version.
>
> ## W3. Benchmark-style illustration beyond the toy problem
> Thank you. As suggested, now **we have adapted the IHDP and ACIC 2016 benchmark datasets** to demonstrate the issue. The **new results** will be added to the paper and are available at: https://sites.google.com/view/oib-holistic. The new results follow the same trend as our simulation and further support the paper’s central point that restricted intervention modelling can induce a value gap relative to holistic optimisation.
>
> IHDP is used to study the impact of child care home visits on the cognitive score of the child; however, it treats prenatal care, alcohol, drugs, and smoking as covariates, which are legitimate actionable factors with potential interactions. This supports our point that, in practice, there are often multiple simultaneous interventions. We therefore adapt IHDP into a multi-treatment benchmark by treating prenatal care, alcohol, drugs, and smoking as interventions, leading to a total of five interventions. Similarly, since the ACIC 2016 covariates were derived from IHDP, we adapted the ACIC 2016 benchmark by considering four interventions, including the one originally included in the dataset. These additions strengthen the practical support for the position.
>
> **Action:** We will add the new results to the camera-ready paper.
>
> ## Q1. Are you aware of benchmark-level problems where the OIB can be estimated?
> The commonly used CATE benchmarks (e.g., Jobs, IHDP, ACIC) are largely structured around single interventions, which is itself part of the paper’s motivation. However, our adaptations of IHDP and ACIC show that OIB can be demonstrated at the benchmark level by treating actionable co-interventions as interventions and comparing holistic and restricted optimisation. We hope these benchmark adaptations help address the reviewer’s question, while also encouraging the creation of benchmarks in which jointly actionable interventions are native to the task rather than retrofitted.
>
> ## ERs
> We thank the reviewer for these editorial remarks. We will address them in the revision if given the opportunity, including the title phrasing, bolded position statement, and box formatting.

---

> > ### Author Rebuttal · Reviewer_xVAa · 2026-04-03
> >
> > I thank the authors for their rebuttal. I maintain my score.

---

### Official Review · Reviewer_pKVX · 2026-03-13

**Significance:** 4
**Argument Clarity:** 4
**Rating:** 4
**Confidence:** 3

**Questions:**

1. **Is the problem really one of awareness?** The paper frames the issue as a research-reality gap. But researchers in clinical settings likely already understand that drugs interact — the reason single-intervention modelling dominates may be primarily practical: insufficient data to cover the joint intervention space, lack of scalable models, and severe positivity violations. If so, the paper's contribution is pointing out something the field already knows but cannot yet solve, which weakens the position's novelty.

2. **How should practitioners act on this paper today?** Given that the paper acknowledges holistic modelling is often infeasible with current data, what should a researcher or clinician actually do differently after reading this paper, beyond adding a disclaimer to their work?

3. **What is the minimum data requirement for holistic modelling to be viable?** The paper does not discuss how large or how diverse a dataset would need to be for joint optimisation over even a modest number of interventions (e.g., 3-4 binary treatments) to be reliable. This would help readers assess feasibility in their own domains.

**Alternative Views Section:**

Yes

**Compliance With Llm Reviewing Policy A Conservative:**

Affirmed.

**Discussion Potential:**

3

**Final Justification:**

The rebuttal is mostly OK. Note that, I have a question "Could you illustrate the results for W1？I don't understand. The blue and orange curves overlapped; is it good? The curves increased as the interaction scale increased; is that what you want to show?" The author has not answered it yet.
Therefore, I maintain my score as 4: Borderline accept.

**Paper Summary:**

This paper argues that current ITE methods, benchmarks, and software pipelines focus on the causal effect of a single intervention. The problem is that, in practice, many interventions interact with each other. Even if the causal effect for each individual intervention is estimated correctly, using it to make personalised treatment decisions can still be suboptimal. Therefore, the field needs holistic  modelling of multiple interventions.

**Position:**

Yes

**Position In Title:**

Yes

**Related Work:**

4

**Strengths And Weaknesses:**

## Strength

1. **Practical problem**. The paper discussed a real gap between how ITE methods are designed (single intervention) and how clinical decisions are actually made (multiple concurrent interventions).
2. **Clear writing and structure**. The paper formalises the problem precisely with OIB, systematically addresses counterarguments in Section 5, and provides actionable recommendations in Section 6, making the logical flow easy to follow.
3. **Formal analysis**. The paper provides a constructive proof (Section 4) with a concrete structural causal model, rather than making the argument purely verbally.
4. **Thorough literature review**. Table 1 offers a well-organised comparison of existing work along multiple relevant dimensions, and Section 5 fairly engages with alternative perspectives.
5. **Simulation support**. The paper includes a simple simulation that empirically corroborates the theoretical claims, showing how the value gap grows with interaction strength and worsens with poor overlap.


## Weakness

1. **No quantitative evidence on real data**. The paper shows that OIB exists both theoretically and in a toy simulation, but it never quantifies the gap on any real-world dataset, leaving the practical significance unclear.
2. **No holistic model demonstrated**. The paper argues for holistic modelling but does not implement one or show it actually improves decisions over restricted approaches.
3. **Shallow engagement with existing holistic approaches**. Existing multi-intervention methods like Chauhan et al. (2025a) are dismissed briefly as "complex" without analysing what specifically makes them inadequate — complexity alone is not a valid critique when the problem itself is combinatorial.

**Support:**

4

---

> ### Author Rebuttal · Authors · 2026-03-27
>
> We are grateful for the reviewer’s time and thoughtful review, and appreciate the recognition of the practical problem, clear writing, formal analysis, thorough literature coverage, simulation support, and concrete actions. We respond to all comments below; we add all items labelled with **Action** in the camera-ready version.
>
> ## W1. No quantitative evidence on real data
>
> Thank you. As suggested, now **we have adapted the IHDP and ACIC 2016 benchmark datasets** to demonstrate the issue. The **new results** are available at: https://sites.google.com/view/oib-holistic. The new results follow the same simulation and further support the paper’s central point that restricted intervention modelling can induce a value gap relative to holistic optimisation.
>
> ## W2. No holistic model demonstrated
>
> We want to clarify that **Appendix C** already implements both the holistic and restricted approaches, with the corresponding code also available as an attachment. _The plotted OIB is precisely the value gap between the policies induced by holistic and restricted intervention modelling. This therefore directly shows the decision-quality difference between them._
>
> **Action:** We will add the new experiments to our camera-ready paper.
>
> ## W3. Shallow engagement with existing holistic approaches
> We apologize that our wording here was too brief. Our intention was not to dismiss emerging multi-intervention work merely because it is complex. Rather, our point is that there is an important emerging body of work moving in the right direction, and we welcome it. At the same time, we still lack methods for multiple simultaneous interventions that explicitly address key challenges such as data scarcity and overlap (see Table 1).
>
> **Action:** We will clarify our wording.
>
> ## Q1. Is the problem really one of awareness?
>
> In our experience, the dominance of restricted intervention modelling is driven by practical challenges, including sparsity, combinatorial growth of the action space, and positivity violations. Our claim is not that the field is simply unaware that interventions interact. Rather, our claim is that these constraints do not remove the conceptual mismatch between restricted-intervention modelling and deployment-time personalised treatment when multiple interventions are actionable, nor do they remove its potential consequences for personalised decision support.
>
> We believe the awareness component still matters because, at present, methods and software pipelines that optimise over restricted intervention sets are often discussed under the broad language of personalised treatment. _Making this mismatch explicit can improve the field in two immediate ways:_
>
> 1. it can lead to clearer reporting and more appropriate scoping of claims in applied work, avoiding harmful consequences by restricting use to the scope of modelling; and
> 2. it can motivate methodology that directly targets the challenges currently forcing restricted modelling.
>
> ## Q2. How should practitioners act on this paper today?
> Thanks. As mentioned in response Q1, explicit reporting of these limitations will directly acknowledge the scope of existing methods or tools for supporting personalised decision making. This will encourage researchers to solve the issues leading to restricted intervention modelling, and appropriate reporting (as per the 'Recommended Reporting Template' box) will avoid unintended consequences.
>
> Upon reading your comment, we realised that we should state this more clearly. Our intended practical message is not only to “add a disclaimer”. Rather, after reading the paper, a practitioner should:
>
> 1. explicitly declare which interventions are actionable at deployment;
> 2. distinguish retrospective effect characterisation from deployment-time decision support;
> 3. avoid interpreting restricted-intervention optimisation as holistic personalised treatment when omitted interventions are actionable; and
> 4. when restriction is unavoidable, report scope limitations and include sensitivity analyses or support diagnostics to assess the risk of OIB.
>
> **Action:** We will revise the above presentation along your suggestion.
>
> ## Q3. What is the minimum data requirement for holistic modelling to be viable?
> Thank you for this important question. We do not think there is a single universal minimum-data threshold, because viability depends on the size and structure of the joint intervention space, overlap, treatment prevalence, interaction complexity, covariate dimension, and the extent to which the model can share information across related intervention combinations. Precisely because of this, we emphasise support diagnostics, overlap-aware methods, restricted recommendation sets, and abstention/deferral when support is insufficient, rather than suggesting that holistic optimisation is always feasible with current data.

---

> > ### Author Rebuttal · Reviewer_pKVX · 2026-04-02
> >
> > I still have a small question. But overall, I remain positive about this paper.

---

### Official Review · Reviewer_9Kzn · 2026-03-23

**Significance:** 2
**Argument Clarity:** 3
**Rating:** 4
**Confidence:** 4

**Questions:**

None

**Alternative Views Section:**

Yes

**Compliance With Llm Reviewing Policy A Conservative:**

Affirmed.

**Discussion Potential:**

2

**Final Justification:**

The author resolved my concern in rebuttal. But I still have some reservations on how significant and original this position is.

**Paper Summary:**

The paper states the position that we need to take a holistic approach of estimating ITE, i.e. incorporating non-restrcied intervention set.

**Position:**

Yes

**Position In Title:**

Yes

**Related Work:**

3

**Strengths And Weaknesses:**

Strength:
clear position, comprehensive altenrative views, and concrete actions

Weakness:
My main concern is the significance and originality of the position. The position itself is not new, basically a paraphrase of the OIB and has been studied before. I think to further support the paper, some recent examples demonstrating why this pisition is even more important as new technology evolves would be better. At the current stage, this is just stating a quite common concensus.

**Support:**

3

---

> ### Author Rebuttal · Authors · 2026-03-27
>
> We are grateful for the reviewer’s time and thoughtful review, and for recognising the paper’s clarity, engagement with alternative views, and concrete call for actions. We respond to all comments below; we add all items labelled with **Action** in the camera-ready version.
>
> > Significance
>
> _Why our topic matters:_ On comparing holistic and restricted intervention modelling, or simply looking at their optimisation equations, it may seem obvious that restricted intervention modelling can lead to worse achievable outcomes and misaligned decisions. However, our paper’s strength is to make explicit why this issue matters for the field: the community is implicitly encouraging restricted intervention modelling for personalised treatments.
>
> _Why our topic is overlooked:_ Much of the CATE research is motivated by personalised treatments, and despite most real-world problems involving multiple simultaneous interventions, much of the existing methodological work, applied research, benchmarks, and software pipelines remain primarily focused on single interventions. As a case in point, a recent overview in JAMA Network Open [1] reviewed 65 ML papers with methods for understanding treatment effects in medicine, but **none** has looked at treatment combinations. This mismatch between deployment-time personalised treatment objectives and restricted intervention modelling is especially consequential in safety-critical healthcare applications, where it can contribute to missed beneficial interventions and adverse outcomes.
>
> _How our Position paper adds:_ Our position formalises the issue of restricted intervention modelling, explains why alternative views are insufficient in the personalised treatment setting, and outlines concrete implications for reporting, evaluation, benchmark design, software pipelines, and future research. In particular, we hope it will encourage (i) more appropriate reporting, especially in applied work, so that claims are restricted to the scope of modelling and unintended consequences are avoided, and (ii) new research that addresses the challenges driving restricted modelling, including overlap, data scarcity, and multi-intervention estimation.
>
> **Action:** We will add the above review paper [1] to highlight that our position has been largely overlooked.
>
> > Originality
>
> To the best of our knowledge, this is the _first_ work in the CATE/ITE personalised decision-support literature to formalise omitted intervention bias (OIB), which is **distinct** from omitted variable bias (OVB), as a value-gap induced by intervention-set restriction, and to prove by constructive example that it can arise even when causal effects for the included interventions are correctly identified. If the reviewer is referring to adjacent ideas in related literatures, we do discuss these in the paper; however, our contribution here is specific to personalised decision support under restricted intervention-set modelling.
>
> Additionally, while our theoretical proofs may seem to establish an obvious point, they are crucial for convincing readers, through a formal mathematical argument, that _OIB can arise even when the restricted estimand is correctly identified_. Hence, the argument is not about complex theoretical proofs, but, following the ideas of the Position track rather than the regular paper track, to present our the paper’s argument in complete and rigoros way. They also make explicit something that is easy to miss in practice: the issue is not ordinary estimation bias, and it does not disappear simply because the restricted estimand is correctly identified. We formalise this under two common restricted-modelling scenarios: B1 (ignores omitted interventions from modelling) and B2 (treats omitted interventions as covariates), and under both the Potential Outcomes and Structural Causal Model frameworks, which also improves accessibility and clarity for readers with different formal preferences.
>
> **Action:** We will make the difference between OIB (ours) vs. OVB (existing in the literature) more explicit to spell out more clearly how our argument is novel.
>
> [1] Selby, J. V., et al (2025). Predictive modeling of heterogeneous treatment effects in RCTs: a scoping review. JAMA network open, 8(7), e2522390.

---

> > ### Author Rebuttal · Reviewer_9Kzn · 2026-04-05
> >
> > Thanks for the clarification and I have changed my score.

---

### Decision · Program_Chairs · 2026-04-30

**Decision:**

Reject

**Comment:**

Even in the gold-standard for clinical causal inference, of randomized clinical trials, we oversimplify into single interventions, ignore most patient details and look only at a small set of covariates, and typically oversimplify to look at one outcome.  While we do some better on these dimensions with observational data, we still typically look at single interventions.  But both good and bad outcomes are usually the product of interactions among many patient features and multiple treatments.  CATE and ITE at least consider far more patient features than in a typical clinical trial, but I agree they usually look at only one treatment.  Thus considering treatment interactions is of great importance.  I don't see major barriers though to considering multiple treatments and allowing them to interact in the same way that patient features may interact.

There was robust discussion between authors and reviewers, and many important issues were raised and addressed, but consensus was not reached.